# Efficacy and safety of Tuina (Chinese Therapeutic Massage) for chronic ankle instability: A systematic review and meta-analysis of randomized controlled trials

Liguo Liu[1], Junqun Huang[2], Tao Li[3], Mingwang Qiu[3], Yanling Huang[3], Zhiyong Fan[3], Shan Wu[3]*, Yanbin Huang[3]*

1 The Second Clinical College of Guangzhou University of Chinese Medicine, Guangzhou, China, 2 Guicheng Community Health Service Center, Foshan, Guangdong, China, 3 The Second Affiliated Hospital of Guangzhou University of Chinese Medicine, Guangzhou, China

* wushan6866@sina.com (SW); drhuangyanbin@163.com (YH)

## Abstract

### Objective

The efficacy of tuina in treating chronic ankle instability (CAI) arouses controversy. Therefore, the present study adopted the meta-analysis to evaluate the effectiveness and safety of tuina in treating CAI and aims to provide high-quality evidence for this promising treatment.

### Methods

We searched eight databases from inception to April 1st, 2024 for randomized controlled trials (RCTs) of tuina treatment for CAI, including PubMed, Web of Science, Embase, Cochrane Library, Wanfang database, China National Knowledge Infrastructure database, and VIP Chinese Science and Technique Journals database. Information was independently extracted and bias risks were evaluated by two researchers. To assess the quality of the studies, we utilized Cochrane Collaboration's tool and the GRADE evaluation system. Meta-analysis was performed using the RevMan5.4 software.

### Results

Thirteen RCTs involving 984 patients were included in this study. The overall methodological quality of the studies was low. The meta-analysis revealed the followings: (1) the clinical effective rate was higher in the treatment group compared to the control group (OR = 6.51, 95% CI [3.76, 11.28]); (2) the treatment group performed better in reducing the Visual Analogue Scale score (MD = −1.59, 95% CI [−2.59, −0.59]); (3) the Baird-Jackson Ankle Score was superior in the treatment group (MD = 8.20, 95% CI [6.37, 10.04]); (4) the improvement in the AOFAS Ankle Hindfoot Scale was

**Data availability statement:** All relevant data are within the paper and its Supporting Information files.

**Funding:** This work was supported by (1) National Natural Science Foundation of China (Grant No. 81874511); (2) Administration of Traditional Chinese Medicine of Guangdong Province, China (Grant No. 2021060400122234240); The funders had no role in study design, data collection and analysis, decision to publish, or preparation of the manuscript.

**Competing interests:** The authors have declared that no competing interests exist.

greater in the treatment group (MD = 14.52, 95% CI [9.81, 19.23]). All differences were statistically significant. Regarding adverse events, there were no significant differences in incidence rates between the groups.

## Conclusions

Tuina is an effective and safe treatment option for CAI, the conclusions are limited by the methodological quality of the included trials. Further high-quality research is needed to confirm these findings and guide clinical practice.

---

## Introduction

An ankle sprain is a frequent sports-related injury, accounting for 16% to 40% of all sports injuries [1]. An acute ankle sprain causes pain and typically results in a temporary period of reduced functioning and disability [2]. Despite the high frequency of ankle sprains, the ideal management is controversial, and a significant percentage of patients sustaining an ankle sprain never fully recover [3], resulting in a large percentage of patients developing chronic ankle instability (CAI) [4,5].CAI may be defined as persistent complaints of pain, swelling, and/or giving way in combination with recurrent sprains for at least 12 months after the initial ankle sprain [6]. Additionally, associations with joint degeneration and osteochondral lesions have been reported over time [7,8]. Adequate diagnosis, treatment, and prevention of injury recurrence could forestall the development of long-term injury-associated symptoms and substantially reduce the associated socioeconomic burden, including costs related to medical treatment, rehabilitation, and lost productivity due to disability [9,10].

Currently, conservative management is the first line for CAI, with surgery interventions considered only as a last resort [11]. These conservative management include oral non-steroidal anti-inflammatory drugs (NSAIDs), bracing or immobilization, Physical factor therapy, and functional rehabilitation training [12,13]. However, NSAIDs can relieve pain and inflammation, but long-term use may cause side effects such as gastrointestinal discomfort and kidney damage [14]. Bracing or immobilization can provide additional mechanical support, but long-term wearing of braces may lead to muscle atrophy, further weakening the stability of the ankle joint [15]; functional rehabilitation training can improve the function and stability of the ankle joint. However, the effect of rehabilitation training depends on the patient's compliance and training intensity. Some patients may find it challenging to adhere to the training, resulting in poor results [16]. Therefore, finding a treatment that can benefit patients more is urgent.

Tuina is a Chinese therapeutic massage based on traditional Chinese medicine (TCM) theories. It integrates modern biomedical theory (biomechanical function, anatomy, pathology, and physiology) with traditional practice [17]. It uses the hands to act on specific parts of the human body surface to regulate the body's physiological and pathological conditions and achieve the purpose of physiotherapy [18]. Several clinical randomized controlled trials (RCTs) [19–21] have shown that massage plays

an important role in CAI. Tuina can relax the tense soft tissues around the ankle joint, improve blood circulation, and promote better recovery of joint function [22,23].

The medical guidelines have previously studied the potential effects of tuina on CAI, and its benefits have been reported [24]. However, the efficacy of tuina in treating CAI arouses controversy. Meta-analysis, the highest level of evidence, can provide a can provide a more convincing conclusion. Therefore, the present study adopted the meta-analysis to evaluate the effectiveness and safety of tuina in treating CAI and aims to provide high-quality evidence for this promising treatment.

## Materials and methods

The protocol for this systematic review was registered at PROSPERO: CRD42024528705.

### Search strategy

A computerized search in the following databases was performed: China National Knowledge Infrastructure(CNKI), Chinese Scientific Journals Full-Text Database(VIP), Wanfang Database, China BioMedical Literature Database(CBM), Cochrane Library, PubMed, Web of Science, and Embase. From the inception of the library to April 1st, 2024 for tuina in treating CAI. Mainly search terms include: "Massage, Manipulation, Tuina, Chronic ankle injury, Chronic Ankle sprain, chronic ankle instability, Randomized Controlled Trial, Randomly". The search strategy was adjusted according to each database. The search strategy for all databases including the search terms and Boolean operators, is detailed in Table 1.

### Inclusion and exclusion criteria

Studies considered for inclusion were required to fulfill the following criteria: (1) Study Type: Only publicly published clinical randomized controlled trials(RCTs) were eligible; (2) Study Subjects: Participants had to be adults aged 18 years or older, with no restrictions based on gender, age beyond the minimum requirement, or case source. Additionally, a precise and consistent diagnosis of CAS was essential [25]; (3) Interventions: The treatment group had to receive either tuina therapy exclusively or in conjunction with other therapies. The control group, on the other hand, could only employ alternative therapies, with tuina being the sole therapeutic variable under investigation. No limitations were imposed on the precise tuina technique, treatment regimen, timing of the procedure, or the duration of the treatment. (4)Outcomes: One of the following needs to be included: clinical effectiveness rate (CER), the AOFAS Ankle Hindfood Scale, Baird-Jackson Ankle Score, and Visual Analogue Scale (VAS).

Studies fitting any of the following criteria were excluded: Duplicate publications or literature for which full text is not available; conference abstracts, reviews, and animal experiments.

**Table 1. The search strategy.**

PubMed:
#1 ((Massage [MeSH Terms]) OR (Manipulation)) OR (Tuina)
#2 (Chronic ankle injury) OR (Chronic Ankle sprain) OR (Chronic Ankle Instability)
#3 ("Randomized Controlled Trial" [Publication Type]) OR (Randomized [Title/Abstract])
#4 #1 and #2 and #3

CNKI:
#1 (SU = 'Tuina' OR SU = 'Anmo' OR SU = 'Shoufa' OR SU = ' Anmoliaofa')
#2 (SU = 'Chenjiuxinghuaiguanjieniushang' OR SU = 'Chenjiuxinghuaiguanjieshunshang' OR SU = 'Manxinghuaiguanjieshunshang' OR SU = 'Manxinghuaiguanjieniushang')
#3 (TI = 'Suijiduizhaoshiyan' OR TI = 'Linchuanshiyan' OR TI = 'Linchuanyanjiu')
#4 #1 AND #2 AND #3

## Outcomes

The primary outcomes is the CER, a continuous variable indicating meaningful change, which is defined as treatment response. According to the degree of improvement in clinical symptoms and functional activities, clinical efficacy is divided into curative, markedly effective, effective, and ineffective. CER is calculated as the total number of the cured, the markedly effective, and the improved divided by the total number of participants. The secondary outcomes encompassed crucial measures such as the AOFAS Ankle Hindfood Scale, Baird-Jackson Ankle Score, VAS, and the incidence of adverse events, among others. VAS is a commonly used tool to measure pain intensity on a scale of 0–10. The Baird-Jackson Ankle Score assesses ankle function and stability. The AOFAS Ankle Hindfoot Scale evaluates ankle and hindfoot function, pain, and alignment.

## Data extraction

The retrieved literature was imported into EndNote, and 2 researchers independently screened the title and abstract content of the literature with reference to the inclusion and exclusion criteria and then read the full text for a second screening. In case of disagreement, a third person made the final decision. Two independent reviewers extracted the following data using a predesigned data collection form: study characteristics (year of publication and first author), age, disease duration, sample size, treatment duration, interventions, primary outcome indicators, follow-up duration, and adverse effects. We documented the entire process using a PRISMA flowchart for clarity and transparency.

## Risk of bias assessments and quality assessment

The quality of the included literature was evaluated using the Cochrane systematic evaluation manual's bias risk assessment tool for RCTs. Two researchers independently assessed each item as "low risk", "unclear risk", or "high risk" based on criteria such as the random allocation method, allocation concealment, blinding of subjects, blinding in outcome assessment, completeness of outcome data, selective reporting, and other potential sources of bias. In case of any disagreement during the evaluation, a third researcher was consulted to reach a consensus. We assessed the quality of the direct and indirect evidence in this study using the grading method recommended by the Grading of Recommendations Assessment, Development and Evaluation system (GRADE) [26].

## Data synthesis and analysis

Review Manager 5.4 was used for meta-analysis. Initially, a heterogeneity test was performed for each study. If $I^2 \leq 50\%$ and $P > 0.1$, it implied insignificant statistical heterogeneity among the included studies, allowing for the application of the fixed effects model; in contrast, if $I^2 > 50\%$ and $P < 0.1$, it signaled statistical heterogeneity within the studies, necessitating the use of the random effects model. In such cases, either a sensitivity analysis was conducted through an article-by-article exclusion approach or a subgroup analysis was performed to pinpoint the source of heterogeneity. The mean difference (MD) was employed for continuous data. Risk ratios (RRs) for dichotomous data were computed, with 95% confidence intervals (CIs) also estimated. $P < 0.05$ was considered statistically significant. Additionally, when the number of studies in a subgroup exceeded 10, a funnel plot was used to assess potential publication bias.

## Results

### Studies selection

Our initial search identified 445 potentially relevant studies. After checking and excluding duplicates, 195 articles remained. Further, after reading the titles and abstracts of the literature and excluding unqualified studies according to the inclusion and exclusion criteria, 72 articles were included after the screening, ultimately leading to the inclusion of 13 studies. The detailed literature screening process and results are presented in Fig 1.

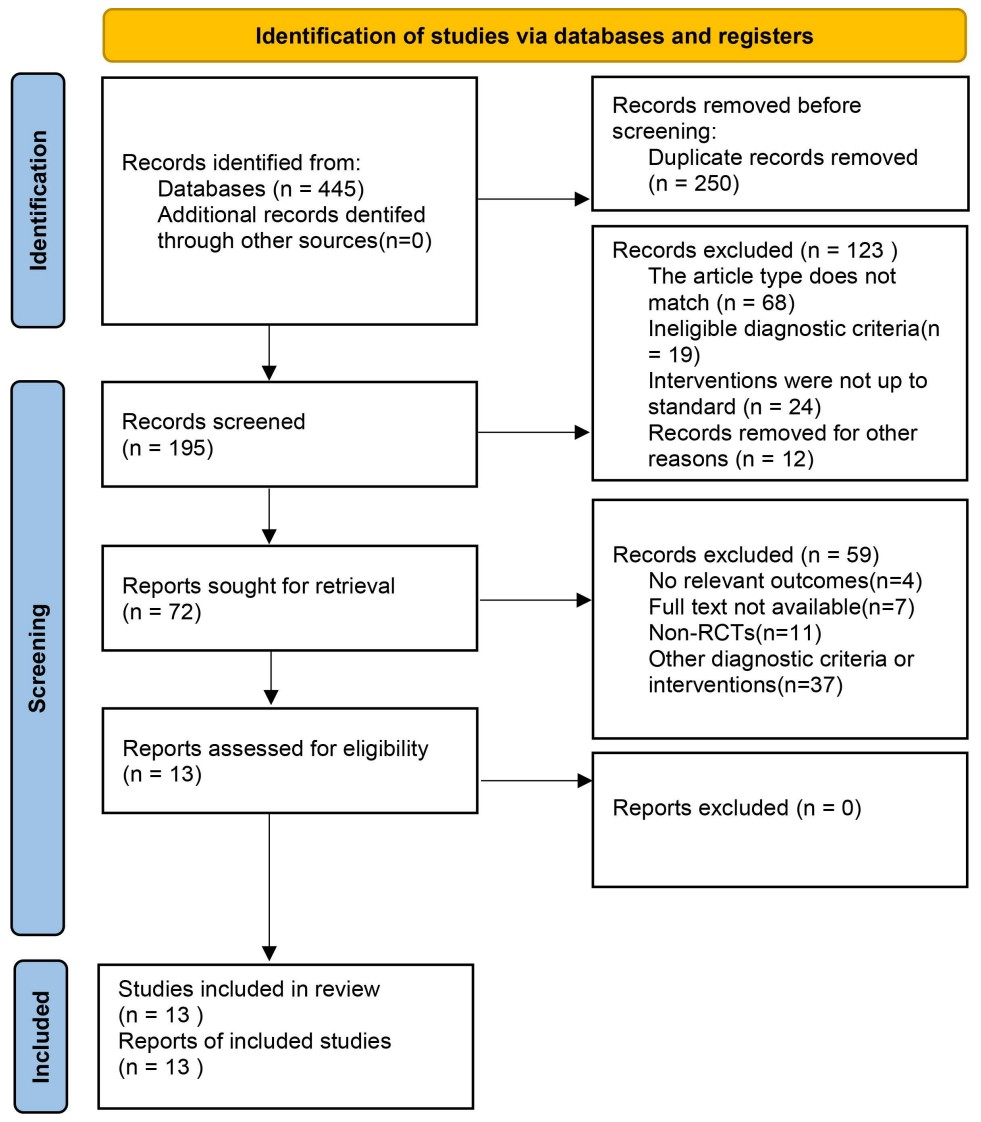

**Fig 1. Flow diagram of literature selection.**

## Study characteristics

Thirteen studies [27–39] with 984 patients were included, of which 502 were in the treatment group and 482 were in the control group (Table 2). Among them, six studies [27–29,31,33,37] used tuina alone as the intervention for the treatment group. Two studies [32,38] combined tuina with the Chinese herbal plaste (CHP). Tuina was used alongside Chinese herbal fumigation (CHF) in 2 studies [30,35]. Three studies [34,36,39] combined tuina with functional rehabilitation training (FRT) for the treatment group. The control groups primarily received CHP, CHF, FRT, and physical factor therapy (PFT). Among the outcome indicators, 10 studies [29–36,38,39] reported VAS, seven studies [27–30,35,37,38] reported Baird-Jackson Ankle Score, three studies [31,33,36] reported AOFAS Ankle Hindfood Scale, eight studies [27,29–32,34,37,38] reported CER.

**Table 2. Basic characteristics of the included studies.**

| Study | Age (y) | | Course of disease (m) | | Sample size | | Interventions | | Treatment duration | Outcomes | Follow-up | Side effects | |
|---|---|---|---|---|---|---|---|---|---|---|---|---|---|
| | T | C | T | C | T | C | T | C | | | Yes or No | T | C |
| Li, 2016 | 34.3±2.3 | 34.5±2.3 | 2.6±1.3 | 2.6±1.2 | 79 | 79 | Tuina | CHF | 2 Weeks | ②, ④ | No | Not stated | Not stated |
| Li, 2012 | 42.97±9.21 | 41.29±8.77 | 7.65±9.43 | 8.47±7.37 | 39 | 37 | Tuina | CHF | 3 Weeks | ② | No | Not stated | Not stated |
| Zhang, 2021 | 40.10±13.91 | 41.31±3.84 | 6.15±7.86 | 6.36±8.95 | 30 | 29 | Tuina | PFT | 2 Weeks | ①, ②, ④ | Yes | 0 | 0 |
| Fan, 2016 | 28.60±8.07 | 28.50±7.16 | 1.05±0.27 | 1.10±0.28 | 30 | 30 | Tuina+CHF | CHF | 2 Weeks | ①, ②, ④ | Yes | 0 | 1 |
| Li, 2017 | 34.63±1.83 | 36.72±2.16 | 13.9±2.83 | 12.1±1.16 | 34 | 32 | Tuina | FRT | 1 Weeks | ①, ③, ④ | Yes | 0 | 0 |
| Liu, 2018 | 31.50±10.97 | 32.27±0.61 | 2.49±0.76 | 2.45±0.68 | 30 | 30 | Tuina+CHP | CHP | 2 Weeks | ①, ④ | No | Not stated | Not stated |
| Chen, 2016 | 36.41±9.84 | 37.13±1.38 | 4.89±1.36 | 4.35±1.24 | 52 | 37 | Tuina | FRT | 2 Weeks | ①, ③ | Yes | 0 | 0 |
| Hong, 2023 | 35.21±2.54 | 35.30±2.48 | 7.12±1.33 | 7.15±1.40 | 25 | 25 | Tuina+FRT | FRT | 2 Weeks | ①, ④ | No | Not stated | Not stated |
| Wu, 2022 | 34.23±4.25 | 33.67±3.83 | 3.51±0.35 | 3.78±0.57 | 48 | 48 | Tuina+CHF | CHF | 2 Weeks | ①, ② | Yes | 0 | 0 |
| Li, 2021 | 29.27±9.44 | 31.67±9.01 | / | / | 30 | 30 | Tuina+FRT | FRT | 4 Weeks | ①, ③ | No | 0 | 0 |
| Lin, 2017 | 35.16±8.33 | 35.73±7.69 | 4.05±1.87 | 4.41±1.83 | 40 | 40 | Tuina | PFT | 2 Weeks | ②, ④ | No | 0 | 0 |
| Shu, 2020 | 40.89±8.83 | 41.81±8.21 | 2.34±1.74 | 2.45±1.61 | 37 | 37 | Tuina+CHP | CHP | 2 Weeks | ①, ②, ④ | No | Not stated | Not stated |
| Gustavo, 2016 | 24.1±2.4 | 24.4±2.4 | / | / | 28 | 28 | Tuina+FRT | FRT | 4 Weeks | ① | Yes | Not stated | Not stated |

**T:** Treatment group. **C:** Control group. **CHF:** Chinese herbal fumigation, an external therapy in Traditional Chinese Medicine that involves using steamed herbal decoctions to bathe or fumigate the body for therapeutic purposes. **PFT:** Physical factor therapy, is a method that uses physical factors such as electricity, light, sound, magnetism, cold, heat, and water to stimulate the human body in order to achieve therapeutic goals. **FRT:** Functional rehabilitation training, refers to therapeutic activities designed to improve or restore specific physical abilities and movements. **CHP:** Chinese herbal plaste, is an external treatment in Traditional Chinese Medicine where medicinal herbs are applied to the skin in the form of patches for healing purposes. ①: Visual Analogue Scale. ②: Baird-Jackson Ankle Score. ③: AOFAS Ankle Hindfood Scale. ④: Clinical Effective Rate.

## Bias risk assessment

The quality of the 13 included studies [27–39] was assessed using the Cochrane Collaboration Network criteria for the risk-of-bias: (1) Randomized allocation method: All included studies were randomized controlled trials, indicating low risk of bias in this domain; (2) Allocation concealment: Four studies [30,33,36,39] had an low risk of bias regarding adequate concealment of participant allocation before randomization, and the rest of the studies were rated as unclear risk of bias; (3) Blinding: Although four studies [30,33,36,39] reported the blinding of participants and personnel, it was not possible to blind of them considering comparisons. Therefore, they were rated as high risk of relevant bias; (4) Completeness of outcome data: All remaining studies maintained consistent sample sizes compared with their initial trials, and outcome data were complete, suggesting low risk of bias; (5) Selective outcome reporting: None of study had missing outcome indicators, so no evidence of selective reporting was found in the remaining studies; (6) Other sources of bias: The available data were not adequate to ascertain the presence of other bias. Figs 2, 3 present a detailed bias risk assessment table generated using RevMan software (Version 5.4, The Cochrane Collaboration, 2020.) for further reference.

## Data synthesis

**CER.** A total of 8 studies [27,29–32,34,37,38] in the included literature reported CER. A heterogeneity test was conducted for CER in both groups across all literature, revealing good homogeneity among the studies ($P=0.91$, $I2=0\%$). Based on these findings, a fixed-effects model was chosen for further analysis. As the inclusion information constituted a dichotomous variable, the Mantel-Haenszel (M-H) method was selected, utilizing OR and its 95% CIs for calculations. The meta-analysis showed that the treatment group demonstrated significantly higher total efficiency compared to the control group (RR=1.22; 95% CI [1.14, 1.30]; $P<0.00001$; Fig 4).

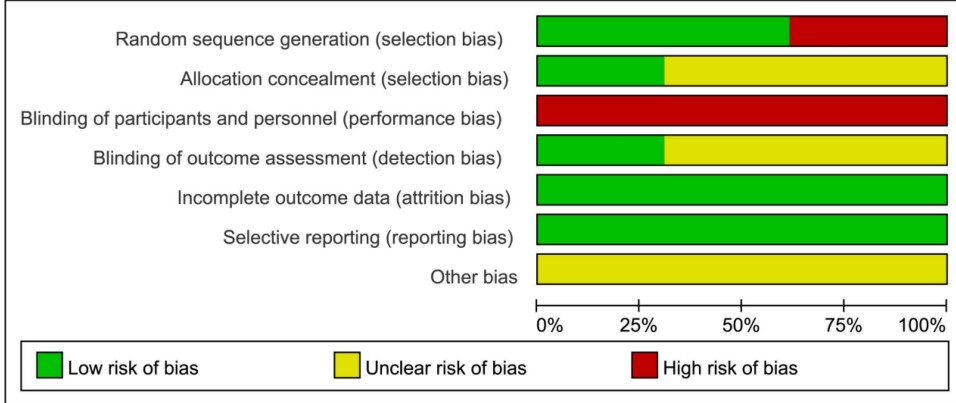

**Fig 2. Risk of bias of all included studies.**

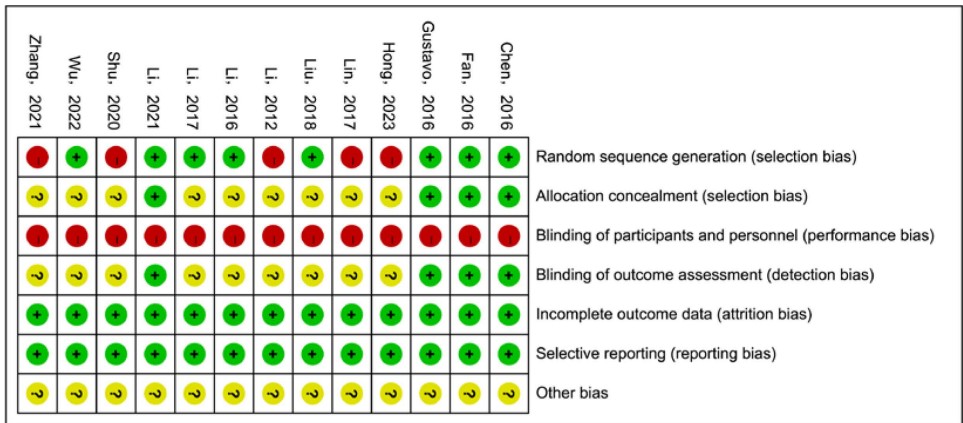

**Fig 3. Risk of bias of all included studies.**

**VAS.** A total of 10 articles [29–36,38,39] have reported VAS. All relevant studies underwent a heterogeneity test, which revealed significant heterogeneity ($P<0.00001$, $I^2=98\%$). Given this heterogeneity, a random effect model was chosen for data analysis since it accommodated the continuous variables involved. Calculations were based on the MD and its 95% CIs. Meta-analysis results suggested that the VAS in the treatment group were notably lower than those in the control group(MD = −1.59; 95% CI [−0.29, −0.59]; $P=0.002$; Fig 5). Given the substantial heterogeneity, a subgroup analysis was performed, based on variations in the adjunctive interventions. The results indicated that the treatment groups significantly outperformed the control group in reducing VAS. These findings are graphically represented in Fig 6.

**Baird-Jackson Ankle score.** A total of 7 papers [27–30,35,37,38] have reported Baird-Jackson Ankle Score. Heterogeneity testing was conducted on the included studies, which revealed significant heterogeneity ($P<0.00001$, $I^2=85\%$). Results suggested that the treatment group outperformed the control group in improving the Baird-Jackson Ankle Score(MD = 8.20; 95% CI [6.57, 10.04]; $P<0.00001$; Fig 7). Given the substantial heterogeneity, sensitivity analyses were conducted to identify its sources. The results indicated that excluding two studies [12,20] significantly reduced the heterogeneity of the remaining literature and the results remain the same as before ($P=0.35$; $I^2=10\%$; Fig 8). The potential reason for the observed differences could be methodological heterogeneity, considering that both studies contrasted tuina with physiotherapy.

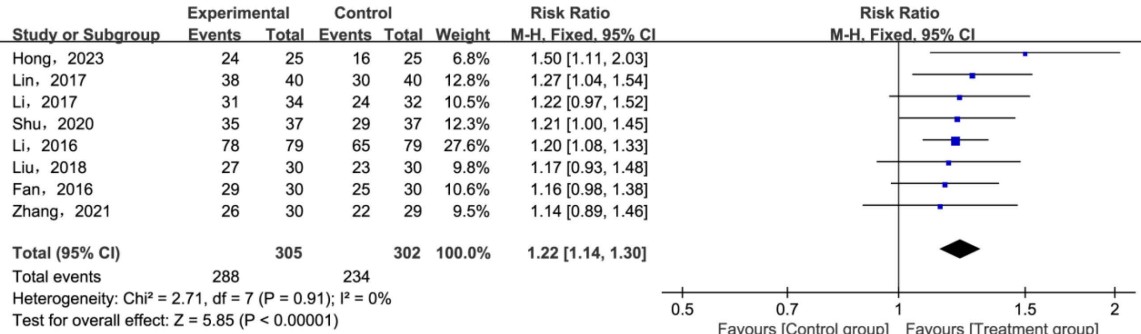

**Fig 4. Forest plot for clinical effectiveness rate (CER).**

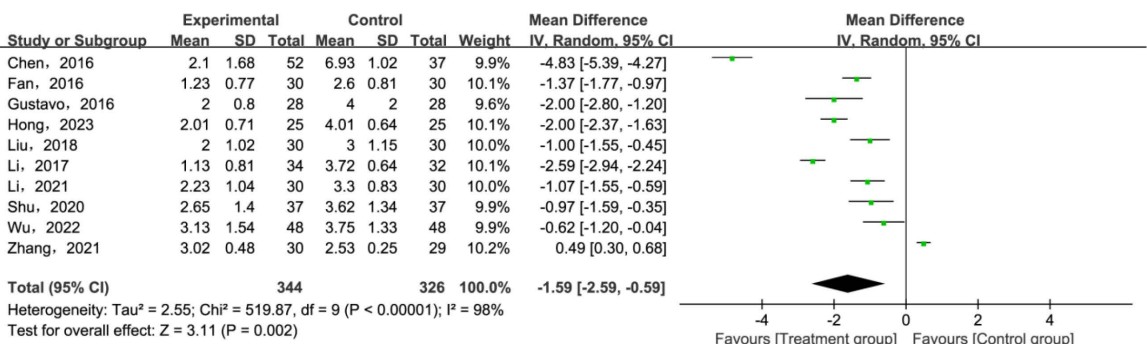

**Fig 5. Forest plot for Visual Analogue Scale (VAS).**

**AOFAS Ankle Hindfood Scale.** Three articles [31,33,36] have reported AOFAS Ankle Hindfoot Scale. All relevant studies underwent a heterogeneity test, which revealed significant heterogeneity ($P < 0.00001$, $I^2 = 93\%$). Consequently, a random effect model was adopted. The results suggested that the treatment group outperformed the control group in terms of improving AOFAS Ankle Hindfoot Scale (MD = 14.52; 95% CI [9.81, 19.23]; $P < 0.00001$; Fig 9). However, the limited number of the studies incorporated led to significant heterogeneity, thereby impairing the reliability of the result.

**Adverse events.** Adverse events were reported in 7 studies [29–31,33,35–37].With the exception of a single article [30] that documented an adverse event within the control group (wherein the individual recovered spontaneously after 48 hours), no other studies mentioned any adverse events throughout the duration of the study.

## Quality assessment

The certainty of evidence was evaluated for CER, VAS, Baird-Jackson Ankle Score, and AOFAS Ankle Hindfood Scale. Our findings indicated that the VAS and AOFAS Ankle Hindfoot Scale provided very low-grade evidence, while the Baird-Jackson Ankle Score provided low-grade evidence. Only the CER provided moderate-grade evidence. Such downgrade was primarily due to the poor methodological quality, high heterogeneity, and insufficient sample size among relevant trials.

## Publication bias

Since the number of RCTs included in this study for all outcomers was less than 10, publication bias could not be evaluated.

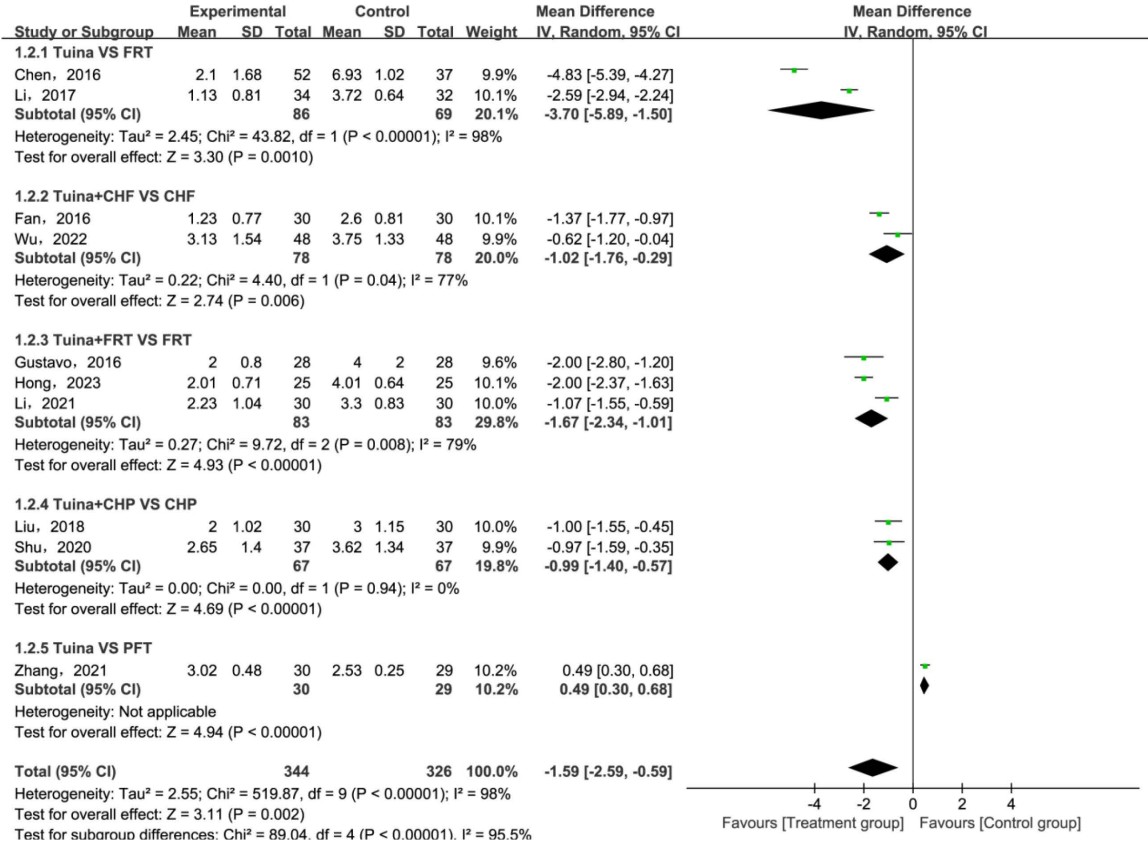

**Fig 6. Forest plot of sub-group analysis for Visual Analogue Scale (VAS).**

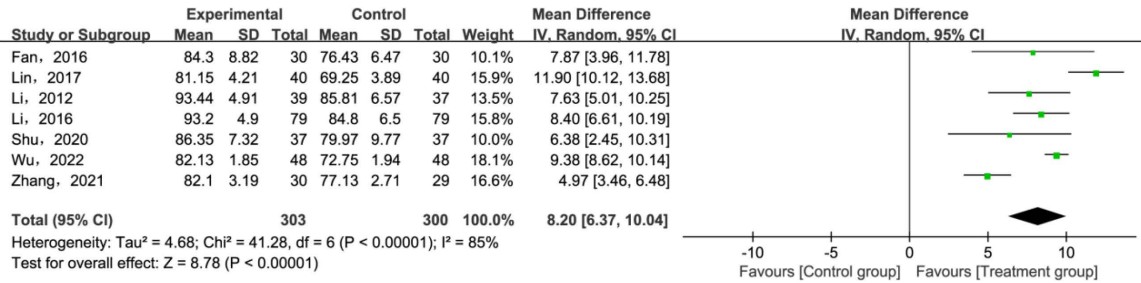

**Fig 7. Forest plot for Baird-Jackson Ankle Score.**

## Discussion

The findings indicate that tuina, either alone or combined with other therapies, effectively alleviated ankle pain and enhanced the ankle joint's range and function of motion in the treatment of CAI.

Moderate-quality evidence revealed that tuina, both as a standalone treatment and when combined with other conservative therapies, has a higher CER than non-tuina approaches. Previous study has also yielded similar results, Loudon [40] concluded that tuina could improve the range of motion of the ankle joint, reduce pain, and enhance the function of subacute/chronic lateral ankle sprains, demonstrating ideal clinical efficacy.

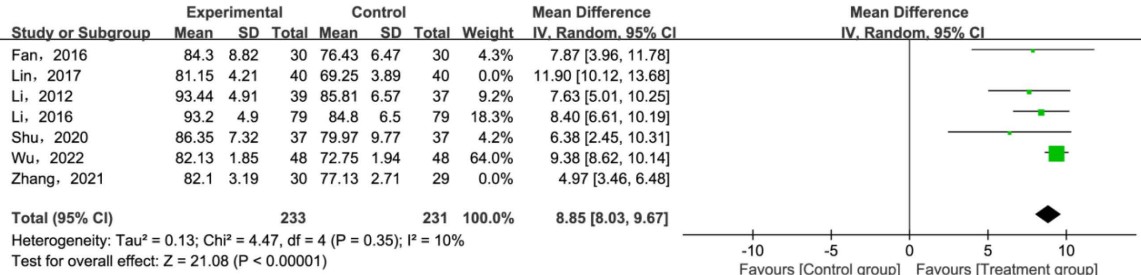

**Fig 8. Forest plot for Baird-Jackson Ankle Score.**

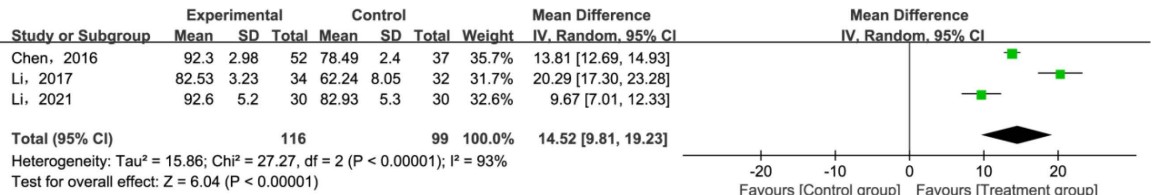

**Fig 9. Forest plot for AOFAS Ankle Hindfood Scale.**

Secondly, the treatment group showed a notable reduction in pain intensity as measured by the VAS. Despite significant heterogeneity in the VAS outcomes, sensitivity analyses suggest that the differences in adjunctive interventions may partly explain this variability. Future studies should detail the specifics of tuina techniques to improve reproducibility and understanding of the intervention's effects. Furthermore, the treatment group demonstrated superior functional outcomes, as evidenced by improved Baird-Jackson Ankle Score and AOFAS Ankle Hindfoot Scale. Previous research supports such findings: Meng [41] demonstrated that tuina could restore the mechanical balance of the ankle joint and aid in repairing damaged ligaments through finite element analysis. Similarly, Chen [42] showed with ultrasound that tuina therapy significantly improves the thickness, width, and dynamic tensile elasticity of specific ankle joint ligaments in patients with CAS. While the evidence for these secondary outcomes is rated as low to very low quality due to methodological limitations and high heterogeneity, they nonetheless contribute to the overall picture of Tuina's efficacy in treating CAI.

The substantial heterogeneity observed in outcomes like the VAS and Baird-Jackson Ankle Score necessitates further exploration into the sources of this variability. The specifics of tuina techniques, including movement type, force, angle, duration, and treatment details such as frequency and overall duration, might have influenced our observed heterogeneity. Documenting these specifics could enhance the assessment of tuina's effects.

Our study reveals that combining tuina and other therapies yields the most favorable clinical effectiveness. Tuina involves the precise readjustment of misaligned ankle joints and restoration of displaced tendons, thereby restoring the injured ankle's anatomical integrity and biomechanical function [43]. This corrective action alleviates pain and discomfort and promotes healing and recovery. Secondly, when combined with other conservative therapies such as functional rehabilitation training, tuina provides a comprehensive treatment approach that addresses the injury's physical and functional aspects. This finding aligns well with previous systematic reviews, particularly the study by Yang [44], which compared various conservative treatment strategies for acute ankle sprains. Their analysis concluded that combining tuina and other therapies yielded the best clinical outcomes.

Despite the positive findings, the suboptimal methodological quality of the included studies limits the reliability of our conclusions. Upon analysis, it became apparent that the quality of evidence suffered primarily due to risks of bias, limited

sample sizes, and significant heterogeneity. These factors variously undermined the quality of the evidence in our study. Deficiencies in methodological design and reporting quality weaken the credibility and authenticity of clinical research, subsequently leading to reduced utilization and conversion rates of research findings [45]. Given tuina's intricate and varying nature, most studies adopted a non-blinded pragmatic approach to assess treatment effectiveness, potentially increasing risk of bias. More rigorous research designs, such as sham-controlled, double-blinded studies, are crucial to minimize bias when evaluating tuina effectiveness. Additionally, most studies conducted only short-term follow-ups or none, with mid to long-term follow-ups being rare. This suggests that the long-term effectiveness of tuina still needs consideration.

No significant adverse safety events were reported in the included studies, suggesting a favorable safety profile for Tuina. However, the sample size may not be sufficient to detect rare but serious side effects. Larger, long-term studies are needed to provide a more robust assessment of Tuina's safety.

Given these limitations, we advocate for more rigorous research designs in future studies, such as double-blinded, sham-controlled RCTs with larger sample sizes and extended follow-up periods. Such studies will help to minimize bias, improve the generalizability of findings, and provide a more definitive assessment of Tuina's efficacy and safety in treating CAI.

In conclusion, while the current study suggests that Tuina is an effective and safe treatment option for CAI, the conclusions are limited by the methodological quality of the included trials. Further high-quality research is needed to confirm these findings and guide clinical practice.

## Conclusion

The findings of this study demonstrated that tuina, either alone or combined with other therapies, effectively alleviated ankle pain and enhanced the ankle joint's range of motion and motor function in the treatment of CAI. This approach exhibited promising clinical efficacy and a high safety profile. However, the overall reliability of this conclusion is limited by the methodological quality of the included trials. As such, the conclusion drawn from this study requires further validation through more rigorous, large-sample RCTs.

## Supporting information

**S1 Table. GRADE evaluation of included studies.**
(PDF)

**S2 Table. Table of all studies identified in the literature search.**
(XLSX)

**S1 File. PRISMA_checklist_2020.**
(DOC)

**S2 File. Included studies.**
(ZIP)

## Author contributions

**Conceptualization:** Liguo Liu, Junqun Huang.

**Funding acquisition:** Shan Wu, Yanbin Huang.

**Investigation:** Tao Li, Zhiyong Fan.

**Methodology:** Mingwang Qiu, Yanling Huang.

**Writing – original draft:** Liguo Liu, Junqun Huang.

**Writing – review & editing:** Shan Wu, Yanbin Huang.

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
