## [Decision Letter · Decision Letter 0]

2 Jan 2025

PONE-D-24-45820Clinical Efficacy and Safety of Manipulation on Chronic Ankle Sprains: A Systematic Review and Meta-AnalysisPLOS ONE

Dear Dr. Huang,

Thank you for submitting your manuscript to PLOS ONE. After careful consideration, we feel that it has merit but does not fully meet PLOS ONE’s publication criteria as it currently stands. Therefore, we invite you to submit a revised version of the manuscript that addresses the points raised during the review process.

We look forward to receiving your revised manuscript.

Kind regards,

Nan Jiang

Academic Editor

PLOS ONE

Journal Requirements:

“This work was supported by (1) National Natural Science Foundation of China (Grant No. 81874511); (2) Administration of Traditional Chinese Medicine of Guangdong Province, China (Grant No. 202106040012234240); “

6. PLOS requires an ORCID iD for the corresponding author in Editorial Manager on papers submitted after December 6th, 2016. Please ensure that you have an ORCID iD and that it is validated in Editorial Manager. To do this, go to ‘Update my Information’ (in the upper left-hand corner of the main menu), and click on the Fetch/Validate link next to the ORCID field. This will take you to the ORCID site and allow you to create a new iD or authenticate a pre-existing iD in Editorial Manager.

7. Please include a copy of Tables 1, 2 and 3 which you refer to in your text on pages 2, 4 and 7.

9. As required by our policy on Data Availability, please ensure your manuscript or supplementary information includes the following: 

Reviewers' comments:

Reviewer's Responses to Questions

**Comments to the Author**

1. Is the manuscript technically sound, and do the data support the conclusions?

Reviewer #1: Yes

Reviewer #2: Partly

2. Has the statistical analysis been performed appropriately and rigorously? 

Reviewer #1: Yes

Reviewer #2: Yes

3. Have the authors made all data underlying the findings in their manuscript fully available?

Reviewer #1: Yes

Reviewer #2: Yes

4. Is the manuscript presented in an intelligible fashion and written in standard English?

Reviewer #1: Yes

Reviewer #2: No

5. Review Comments to the Author

Reviewer #1: The manuscript provides a systematic review and meta-analysis evaluating the efficacy and safety of manipulation techniques for chronic ankle sprains (CAS). The authors identified 13 studies with a total of 984 cases and performed robust statistical analyses using established tools such as RevMan 5.4 and GRADE. The study addresses a relevant topic with potential clinical impact. However, several areas require clarification and improvement before the manuscript can be considered for publication as following:

- While the authors mention the low methodological quality of the included studies, a more detailed discussion on how this impacts the findings would strengthen the conclusions. Suggestions for mitigating these limitations in future research should also be included. For example, advocating for double-blinded, sham-controlled RCTs with larger sample sizes.

- The substantial heterogeneity in outcomes like the Visual Analogue Scale (VAS) and Baird-Jackson Ankle Score is acknowledged. However, further exploration into the sources of this variability (e.g., differences in manipulation techniques or adjunctive therapies) is needed. This could improve the robustness of the conclusions.

- The manuscript states that the certainty of evidence was low or very low for most outcomes. Including a table summarizing the GRADE results for each outcome would improve the presentation and allow readers to understand the evidence hierarchy better.

- Although no significant adverse events were reported, it would be valuable to discuss whether the sample size is sufficient to detect rare but serious side effects. This could strengthen the statement regarding safety.

- The search strategy and inclusion/exclusion criteria are well described. However, the authors should provide the full search string in a supplementary table to ensure reproducibility.

- The manuscript notes that publication bias could not be evaluated due to the small number of studies. A funnel plot or sensitivity analysis (even with limitations) might provide additional insights.

- The manuscript contains minor grammatical errors and awkward phrasing. For example, "timely and appropriate intervention can effectively reduce the long-term consequences" could be rephrased for clarity. also, in introduction 5th line (long term) was duplicated.

- The figures (e.g., forest plots) are informative but require clearer labeling to enhance readability. The PRISMA flowchart is helpful but should specify reasons for exclusion more succinctly. Tables 1,2 &3 were missed in the pdf file, so I can not review tables.

- The meta-analytic outcomes for the primary and secondary measures are well presented but could be enhanced by briefly summarizing them in the discussion section.

Reviewer #2: Introduction

o The study does not provide a clear definition of chronic ankle sprain. Does this refer to chronic ankle instability? Is it ethical to perform manipulation on ankle sprains, which are typically in the inflammatory stage? What is the intended purpose of manipulation in these cases? If the goal is to improve range of motion, why is this technique being applied to ankle sprains in the inflammatory stage?

o In the sentence of “if left untreated, mistreated, or excercise improperly, the ankle joint may develop continuous localized swelling and scar tissue formation, leading to long term long-term pain, limited mobility, and eventually, chronic ankle sprain (CAS)”, correct the typo 'excercise' to 'exercise' and revise the sentence to eliminate the repeated phrase 'long term long-term.'

o You might elaborate briefly on the benefits or limitations of each conservative treatment mentioned to provide a more balanced overview.

o Clarify what specific benefits manipulation offers compared to other treatments and whether it is supported by modern medical guidelines.

o In the sentence of “Although several clinical studies have evaluated the effectiveness of manipulation in improving CAS, there is still a lack of high-quality evidence to convince the broader medical community of its efficacy, suggest specifying what 'high-quality evidence' entails, such as randomized controlled trials, systematic reviews, or meta-analyses.

Method

o Are there limited keywords used in search strategies?

o Provide the syntax used.

Result - Study Characteristics

o Clarify what "manipulation" entails. Is it a specific manual therapy technique? Defining this term will enhance understanding.

o Provide more context or examples of the "external application of traditional Chinese medicine" for readers unfamiliar with these methods.

o "Among the outcome indicators, 10 studies...reported VAS", expand on what VAS, Baird-Jackson Ankle Score, AOFAS Ankle Hindfoot Scale, and CER measure. Brief descriptions will help readers interpret the findings.

o Provide explanations about control groups such as herbal fumigation, herbal compress.

o What is the difference between functional exercise and physical therapy?

o Correct the typo "hherapy" to "therapy."

o Consider restructuring the sentence for smoother readability, such as: "In three studies [17,19,22], the treatment group received manipulation combined with functional exercises."

Bias Risk Assessment

o In the sentence of "Four studies...reported lost visit data, but all offered clear explanations for the losses and their treatment.", clarify whether the treatment of lost visit data (e.g., imputation methods) was appropriate and how it might impact the results.

o In the sentence of "thereby elevating the potential for performance bias.", consider rephrasing for smoother flow, e.g., "which may have increased the risk of performance bias."

Discussion

o Consider restructuring the opening sentence for better clarity and impact. For example, directly state the key findings instead of introducing the analysis.

o In the sentence of "Moderate-quality evidence revealed that manipulation, both as a standalone treatment and when combined with other conservative therapies...", clarify the meaning of "moderate-quality evidence" by referencing the criteria used (e.g., GRADE).

o In the sentence of "This finding aligns well with previous systematic reviews, particularly the study conducted by Yang [23]..., provide a brief summary of Yang's study to contextualize its relevance.

o In the sentence of "Manipulation could correctly readjust misaligned ankle joints and restore displaced tendons and tissues...", ensure that this claim is adequately supported by referenced studies or data. If speculative, indicate as such.

o In the sentence of "Although rated as poor quality evidence, our systematic reviews suggested that manipulation...", clarify why this evidence is rated as "poor quality" and how this impacts the strength of the conclusions.

o In the sentence of "The specifics of manipulation techniques, including movement type, force, angle, and duration...", suggest that future studies document these parameters in detail to improve reproducibility and understanding of the intervention's effects.

o In the sentence of "The primary limitation of this review lies in the suboptimal methodological quality of the included studies.", suggest adding a specific example of how this limitation affected the conclusions, such as the impact of missing data or lack of blinding.

o In this sentence of "This corrective action not only alleviates pain and discomfort but also promotes healing and recovery.", consider breaking up longer sentences for improved readability.

6. PLOS authors have the option to publish the peer review history of their article (what does this mean? ). If published, this will include your full peer review and any attached files.

**Do you want your identity to be public for this peer review?** For information about this choice, including consent withdrawal, please see our Privacy Policy .

Reviewer #1: No

Reviewer #2: No

---

## [Author Response · Author response to Decision Letter 1]

24 Feb 2025

Dear Editors and Reviewers:

Thank you for your letter and for the reviewers’ comments concerning our manuscript (ID: PONE-D-24-45820). Those comments are all valuable and very helpful for revising and improving our paper, as well as the important guiding significance to our researches. We have studied comments carefully and have made correction which we hope meet with approval.

The main corrections in the paper and the responds to the reviewer’s comments are as flowing:

Responds to the academic editor:

1.Please ensure that your manuscript meets PLOS ONE's style requirements, including those for file naming.

Response: The manuscript had been revised in accordance with the requirements of the journal.

2.We note that the grant information you provided in the ‘Funding Information’ and ‘Financial Disclosure’ sections do not match.

Response: Thanks for the suggestion, we've revised this section to make sure they're consistent.

3.Thank you for stating the following financial disclosure: “The author(s) received no specific funding for this work.”At this time, please address the following queries:

Response: The funders had no role in study design, data collection and analysis, decision to publish, or preparation of the manuscript

4.We note that you have provided funding information that is not currently declared in your Funding Statement. However, funding information should not appear in the Acknowledgments section or other areas of your manuscript.

Response: Thanks for your suggestion, We have revised this section.

5.When completing the data availability statement of the submission form, you indicated that you will make your data available on acceptance. We strongly recommend all authors decide on a data sharing plan before acceptance, as the process can be lengthy and hold up publication timelines.

Response: Thanks for the suggestion, we'll take it.

6.Please ensure that you have an ORCID iD and that it is validated in Editorial Manager.

Response: We will update the information to ensure that the corresponding author has the ORCID iD.

7.Please include a copy of Tables 1, 2 and 3 which you refer to in your text on pages 2, 4 and 7.

Response: We had submitted these Supporting materials.

8.Please include captions for your Supporting Information files at the end of your manuscript, and update any in-text citations to match accordingly.

Response: Thanks for the suggestion, we've revised this section.

9.As required by our policy on Data Availability, please ensure your manuscript or supplementary information includes the following:

Response: We submitted the table which contains all the studies that were included after the removal of duplicates and lists the reasons for the exclusion or inclusion of each piece of literature. Additionally, we had submitted a table for the extraction of data for the study, which contains all the data needed for this study to be analyzed. No unpublished studies were cited in this review and all included studies were publicly published. All data were extracted directly from the publicly available full texts of the included studies and No additional data sources were utilized.

Responds to the reviewer’s comments:

Reviewer #1:

1.While the authors mention the low methodological quality of the included studies, a more detailed discussion on how this impacts the findings would strengthen the conclusions. Suggestions for mitigating these limitations in future research should also be included. For example, advocating for double-blinded, sham-controlled RCTs with larger sample sizes.

Response: We have provided a more detailed discussion on how the low methodological quality of the included studies impacts our findings: “Given these limitations, we advocate for more rigorous research designs in future studies, such as double-blinded, sham-controlled RCTs with larger sample sizes and extended follow-up periods. Such studies will help to minimize bias, improve the generalizability of findings, and provide a more definitive assessment of Tuina's efficacy and safety in treating CAI.”

2.The substantial heterogeneity in outcomes like the Visual Analogue Scale (VAS) and Baird-Jackson Ankle Score is acknowledged. However, further exploration into the sources of this variability (e.g., differences in manipulation techniques or adjunctive therapies) is needed. This could improve the robustness of the conclusions.

Response: Thanks for the suggestion, we've revised this section: “The substantial heterogeneity observed in outcomes like the VAS and Baird-Jackson Ankle Score necessitates further exploration into the sources of this variability. The specifics of tuina techniques, including movement type, force, angle, duration, and treatment details such as frequency and overall duration, might have influenced our observed heterogeneity. Documenting these specifics could enhance the assessment of tuina's effects."

3.The manuscript states that the certainty of evidence was low or very low for most outcomes. Including a table summarizing the GRADE results for each outcome would improve the presentation and allow readers to understand the evidence hierarchy better.

Response: Thanks for the suggestion, we have included a table summarizing the GRADE results for each outcome to improve the presentation and clarity of our evidence hierarchy. This table provides readers with a clear understanding of the quality of evidence supporting our findings.

4.Although no significant adverse events were reported, it would be valuable to discuss whether the sample size is sufficient to detect rare but serious side effects. This could strengthen the statement regarding safety.

Response: Thanks for the suggestion, we've revised this section: “No significant adverse safety events were reported in the included studies, suggesting a favorable safety profile for Tuina. However, the sample size may not be sufficient to detect rare but serious side effects. Larger, long-term studies are needed to provide a more robust assessment of Tuina's safety.”

5.The search strategy and inclusion/exclusion criteria are well described. However, the authors should provide the full search string in a supplementary table to ensure reproducibility.

Response: Thanks for the suggestion, we have provided the full search string used in PubMed in a supplementary table to ensure the reproducibility of our study. This search string has been adapted for use in other databases as necessary.

6.The manuscript notes that publication bias could not be evaluated due to the small number of studies. A funnel plot or sensitivity analysis (even with limitations) might provide additional insights.

Response: Due to the small number of included studies, we were unable to conduct a funnel plot to assess publication bias. However, we have performed a sensitivity analysis to explore sources of heterogeneity and reduce the impact of bias on our results. In future studies with a larger number of included studies, we will consider using a funnel plot to assess publication bias in conjunction with sensitivity analysis.

7.The manuscript contains minor grammatical errors and awkward phrasing. For example, "timely and appropriate intervention can effectively reduce the long-term consequences" could be rephrased for clarity. Also, in the introduction, the word "long-term" was duplicated.

Response: Thanks for the suggestion, we have carefully reviewed and corrected grammatical errors and awkward phrasing throughout the manuscript to ensure clarity and accuracy.

8.The figures (e.g., forest plots) are informative but require clearer labeling to enhance readability. The PRISMA flowchart is helpful but should specify reasons for exclusion more succinctly. Tables 1, 2, & 3 were missed in the pdf file, so I cannot review tables.

Response: We have improved the labeling of figures to enhance readability and provided some reasons for exclusion in the PRISMA flowchart. Additionally, I have included missing tables (e.g., Table S3, GRADE evaluation table) to provide comprehensive study information and evidence hierarchy assessments.

9.The meta-analytic outcomes for the primary and secondary measures are well presented but could be enhanced by briefly summarizing them in the discussion section.

Response: Thanks for the suggestion, we've revised this section: “In conclusion, while the current study suggests that Tuina is an effective and safe treatment option for CAI, the conclusions are limited by the methodological quality of the included trials. “

Reviewer #2:

Introduction

1.The study does not provide a clear definition of chronic ankle sprain. Does this refer to chronic ankle instability? Is it ethical to perform manipulation on ankle sprains, which are typically in the inflammatory stage? What is the intended purpose of manipulation in these cases? If the goal is to improve range of motion, why is this technique being applied to ankle sprains in the inflammatory stage?

Response: By "chronic ankle sprain," we indeed refer to chronic ankle instability (CAI), a condition characterized by recurrent instability and pain in the ankle joint. Regarding the ethical considerations, Tuina, as a Chinese Therapeutic Massage, is commonly used in the treatment of musculoskeletal conditions, including ankle sprains. In the inflammatory stage, tuina is applied with caution and under professional guidance to avoid exacerbating inflammation. However, it can be considered in the subacute or chronic stages to restore anatomical alignment, improve range of motion, and reduce pain and disability.

2.In the sentence "if left untreated, mistreated, or excercise improperly, the ankle joint may develop continuous localized swelling and scar tissue formation, leading to long term long-term pain, limited mobility, and eventually, chronic ankle sprain (CAS)", correct the typo 'excercise' to 'exercise' and revise the sentence to eliminate the repeated phrase 'long term long-term.'

Response: Thanks for your suggestion, we have revised this section.

3.You might elaborate briefly on the benefits or limitations of each conservative treatment mentioned to provide a more balanced overview.

Response: Thanks for your suggestion, we have revised this section: “NSAIDs can relieve pain and inflammation, but long-term use may cause side effects such as gastrointestinal discomfort and kidney damage. Bracing or immobilization can provide additional mechanical support, but long-term wearing of braces may lead to muscle atrophy, further weakening the stability of the ankle joint ; functional rehabilitation training can improve the function and stability of the ankle joint. However, the effect of rehabilitation training depends on the patient's compliance and training intensity. Some patients may find it challenging to adhere to the training, resulting in poor results. “

4.Clarify what specific benefits manipulation offers compared to other treatments and whether it is supported by modern medical guidelines.

Response: Tuina is a Chinese therapeutic massage based on traditional Chinese medicine (TCM) theories. Several clinical randomized controlled trials (RCTs) have shown that massage plays an important role in CAI. Tuina can relax the tense soft tissues around the ankle joint, improve blood circulation, and promote better recovery of joint function . In addition, the medical guidelines have previously studied the potential effects of tuina on CAI, and its benefits have been reported.

5.In the sentence of “Although several clinical studies have evaluated the effectiveness of manipulation in improving CAS, there is still a lack of high-quality evidence to convince the broader medical community of its efficacy, suggest specifying what 'high-quality evidence' entails, such as randomized controlled trials, systematic reviews, or meta-analyses.

Response: We agree that high-quality evidence is crucial for validating the efficacy of tuina in improving CAI. Meta-analysis, the highest level of evidence, can provide a can provide a more convincing conclusion. Therefore, the present study adopted the meta-analysis to evaluate the effectiveness and safety of tuina in treating CAI and aims to provide high-quality evidence for this promising treatment.

Method

1.Are there limited keywords used in search strategies?

2.Provide the syntax used.

Response: We have reviewed our search strategies and found that we used a comprehensive set of keywords related to chronic ankle instability, manipulation, and conservative treatments. The syntax used in our search queries varied depending on the database searched, but typically included Boolean operators (AND, OR, NOT) to combine and refine search terms. We have provided more details on our search strategies and syntax in the revised paper.

Result - Study Characteristics

1.Clarify what "manipulation" entails. Is it a specific manual therapy technique? Defining this term will enhance understanding.

2. Provide more context or examples of the "external application of traditional Chinese medicine" for readers unfamiliar with these methods.

Response: We have noticed that the term "manipulation" is misleading and have replaced it with "tuina." Tuina is a Chinese therapeutic massage based on traditional Chinese medicine (TCM) theories. It integrates modern biomedical theory (biomechanical function, anatomy, pathology, and physiology) with traditional practice. It uses the hands to act on specific parts of the human body surface to regulate the body's physiological and pathological conditions and achieve the purpose of physiotherapy.

3."Among the outcome indicators, 10 studies...reported VAS", expand on what VAS, Baird-Jackson Ankle Score, AOFAS Ankle Hindfoot Scale, and CER measure. Brief descriptions will help readers interpret the findings.

Response: Thanks for your suggestion, we have revised this section: “VAS (Visual Analog Scale) is a commonly used tool to measure pain intensity on a scale of 0-10. The Baird-Jackson Ankle Score assesses ankle function and stability. The AOFAS Ankle Hindfoot Scale evaluates ankle and hindfoot function, pain, and alignment. CER (Clinical Effectiveness Ratio) is a measure of treatment effectiveness, typically calculated as the ratio of the number of patients achieving a specific outcome to the total number of patients treated. We have provided brief descriptions of these outcome indicators in the revised paper to aid reader interpretation.”

4.Provide explanations about control groups such as herbal fumigation, herbal compress.

5.What is the difference between functional exercise and physical therapy?

Response: Thanks for your suggestion, we have provided brief descriptions of these terms in the revised paper to aid reader interpretation: “Chinese herbal fumigation (CHF), an external therapy in Traditional Chinese Medicine that involves using steamed herbal decoctions to bathe or fumigate the body for therapeutic purposes. Physical factor therapy (PFT), is a method that uses physical factors such as electricity, light, sound, magnetism, cold, heat, and water to stimulate the human body in order to achieve therapeutic goals. Functional rehabilitation training (FRT), refers to therapeutic activities designed to improve or restore specific physical abilities and movements. Chinese herbal plaste (CHP), is an external treatment in Traditional Chinese Medicine where medicinal herbs are applied to the skin in the form of patches for healing purposes. ”

6.Correct the typo "hherapy" to "therapy."

7.Consider restructuring the sentence for smoother readability, such as: "In three studies [17,19,22], the treatment group received manipulation combined with functional exercises."

Response: Thanks for your suggestion, We have revised these sections.

Bias Risk Assessment

1.In the sentence of "Four studies...reported lost visit data, but all offered clear explanations for the losses and their treatment.", clarify whether the treatment of lost visit data (e.g., imputation methods) was appropriate and how it might impact the results.

Response: Regarding the treatment of lost visit data, we relied on the explanations provided by the original studies and assumed that imputation methods, if used, were appropriate for the context. However, we acknowledge that handling missing data can i

---

## [Decision Letter · Decision Letter 1]

11 Mar 2025

Efficacy and safety of Tuina (Chinese Therapeutic Massage) for Chronic Ankle Instability: A Systematic Review and Meta-Analysis of randomized controlled trials

PONE-D-24-45820R1

Dear Dr. Huang,

We’re pleased to inform you that your manuscript has been judged scientifically suitable for publication and will be formally accepted for publication once it meets all outstanding technical requirements.

Kind regards,

Nan Jiang

Academic Editor

PLOS ONE

Reviewer's Responses to Questions

**Comments to the Author**

1. If the authors have adequately addressed your comments raised in a previous round of review and you feel that this manuscript is now acceptable for publication, you may indicate that here to bypass the “Comments to the Author” section, enter your conflict of interest statement in the “Confidential to Editor” section, and submit your "Accept" recommendation.

Reviewer #2: All comments have been addressed

2. Is the manuscript technically sound, and do the data support the conclusions?

Reviewer #2: Yes

3. Has the statistical analysis been performed appropriately and rigorously? 

Reviewer #2: Yes

4. Have the authors made all data underlying the findings in their manuscript fully available?

Reviewer #2: Yes

5. Is the manuscript presented in an intelligible fashion and written in standard English?

Reviewer #2: Yes

6. Review Comments to the Author

Reviewer #2: Thank you for addressing my comments effectively. I appreciate the thoughtful revisions, which have significantly improved the clarity and quality of the manuscript. The changes have been well implemented, and I have no further concerns.

7. PLOS authors have the option to publish the peer review history of their article (what does this mean? ). If published, this will include your full peer review and any attached files.

**Do you want your identity to be public for this peer review?** For information about this choice, including consent withdrawal, please see our Privacy Policy .

Reviewer #2: No

---

## [Editor Report · Acceptance letter]

PONE-D-24-45820R1

PLOS ONE

Dear Dr. Huang,

I'm pleased to inform you that your manuscript has been deemed suitable for publication in PLOS ONE. Congratulations! Your manuscript is now being handed over to our production team.

Kind regards,

on behalf of

Dr. Nan Jiang

Academic Editor

PLOS ONE